# Effectiveness of a multifaceted prevention programme for melioidosis in diabetics (PREMEL): A stepped-wedge cluster-randomised controlled trial

**Pornpan Suntornsut**[1], **Prapit Teparrukkul**[2], **Gumphol Wongsuvan**[1],
**Wipada Chaowagul**[2], **Susan Michie**[3], **Nicholas P. J. Day**[1,4],
**Direk Limmathurotsakul**[1,4,5]*

**1** Mahidol Oxford Tropical Medicine Research Unit, Faculty of Tropical Medicine, Mahidol University,
Bangkok, Thailand, **2** Department of Medicine, Sunpasitthiprasong Hospital, Ubon Ratchathani, Thailand,
**3** Centre for Behaviour Change and Department of Clinical, Educational and Health Psychology, University
College of London, London, United Kingdom, **4** Centre for Tropical Medicine and Global Health, Nuffield
Department of Medicine, University of Oxford, Oxford, United Kingdom, **5** Department of Tropical Hygiene,
Faculty of Tropical Medicine, Mahidol University, Bangkok, Thailand

* direk@tropmedres.ac

Singapore, SINGAPORE

**Data Availability Statement:** Data cannot be
shared publicly because of the informed consent
did not include the permission to make anonymous

## Abstract

### Background

Melioidosis, an often-fatal infectious disease caused by the environmental Gram-negative
bacillus *Burkholderia pseudomallei*, is endemic in tropical countries. Diabetes mellitus and
environmental exposure are important risk factors for melioidosis acquisition. We aim to
evaluate the effectiveness of a multifaceted prevention programme for melioidosis in diabet-
ics in northeast Thailand.

### Methodology/Principal findings

From April 2014 to December 2018, we conducted a stepped-wedge cluster-randomized
controlled behaviour change trial in 116 primary care units (PCUs) in Ubon Ratchathani
province, northeast Thailand. The intervention was a behavioural support group session to
help diabetic patients adopt recommended behaviours, including wearing rubber boots and
drinking boiled water. We randomly allocated the PCUs to receive the intervention starting
in March 2016, 2017 and 2018. All diabetic patients were contacted by phone yearly, and
the final follow-up was December 2018. Two primary outcomes were hospital admissions
involving infectious diseases and culture-confirmed melioidosis. Of 9,056 diabetics enrolled,
6,544 (72%) received a behavioural support group session. During 38,457 person-years of
follow-up, we observed 2,195 (24%) patients having 3,335 hospital admissions involved
infectious diseases, 80 (0.8%) melioidosis, and 485 (5%) deaths. In the intention-to-treat
analysis, implementation of the intervention was not associated with primary outcomes. In
the per-protocol analysis, patients who received a behavioural support group session had
lower incidence rates of hospital admissions involving infectious diseases (incidence rate
ratio [IRR] 0.89; 95%CI 0.80–0.99, p = 0.03) and of all-cause mortality (IRR 0.54; 95%CI

data being shared publicly. Data are available from Mahidol-Oxford Tropical Medicine Research Unit Data Access Committee (contact via Phaikyeong Cheah [Phaikyeong@tropmedres.ac]) for researchers who meet the criteria for access to confidential data.

**Funding:** The study was funded in whole, or in part, by the Wellcome Trust (Grant 101103/Z/13/Z to DL). For the purpose of Open Access, the author has applied a CC BY public copyright licence to any Author Accepted Manuscript version arising from this submission. The funders had no role in study design, data collection and analysis, decision to publish, or preparation of the manuscript.

**Competing interests:** The authors have declared that no competing interests exist.

0.43–0.68, p<0.001). However, the incidence rate of culture-confirmed melioidosis was not significantly lower (IRR 0.96, 95%CI 0.46–1.99, p = 0.66).

## Conclusions/Significance

Clear benefits of this multifaceted prevention programme for melioidosis were not observed. More compelling invitations for the intervention, modification of or addition to the behaviour change techniques used, and more frequent intervention may be needed.

## Trial registration

This trial is registered with ClinicalTrials.gov, number NCT02089152.

### Author summary

Melioidosis, an infectious disease caused by environmental bacterium *Burkholderia pseudomallei*, is endemic in tropical countries. Diabetes mellitus is the most important risk factor, and routes of infection include skin inoculation, ingestion and inhalation. Prevention guidelines recommend that residents, rice farmers and visitors should wear protective gear such as rubber boots when in direct contact with soil and environmental water, and consume only boiled or bottled water. Here, we conducted a cluster randomized controlled trial to evaluate effectiveness of a multifaceted prevention programme for melioidosis in diabetic patients in northeast Thailand. We enrolled 9,056 diabetic patients in 2014. We randomly allocated primary care units as the unit of randomization to receive the intervention starting in March 2016, 2017 and 2018. All diabetic patients were contacted by phone yearly. We found that diabetic patients who received a behavioural support group session had lower incidence rates of hospital admissions involving infectious diseases and of all-cause mortality, but not of culture-confirmed melioidosis. In conclusion, clear benefits of this multifaceted prevention programme for melioidosis were not seen. We propose that more compelling invitations for the intervention, modification of or addition to the behaviour change techniques used, and more frequent intervention may be needed.

## Introduction

Melioidosis is an often-fatal infection caused by the environmental Gram-negative bacillus *Burkholderia pseudomallei*, found in soil and water. The disease is considered highly endemic [1–3] and an increasing number of melioidosis cases are increasingly reported in other tropical regions including South Asia, Africa and Central and South America [4–7]. A recent modelling study estimated that there were 165,000 melioidosis cases per year worldwide, of which 54% die [8]. Diabetes mellitus is the most important risk factor for melioidosis [1–3]. About half of melioidosis patients have underlying diabetes, and diabetic patients have a 12-fold higher risk of melioidosis after adjusting for age, sex and other risk factors [9,10]. Skin inoculation, ingestion and inhalation are all important routes of infection from environmental *B. pseudomallei* [1]. Patients commonly present with sepsis and septic shock with or without localized or disseminated organ involvement such as pneumonia, urinary tract infection, and

central nervous system infection [1–3]. Culture positive for *B. pseudomallei* from any clinical specimens is the gold standard for diagnosis [1].

There is a strong need for interventions to prevent melioidosis with proven effectiveness, particularly for high-risk populations such as diabetic patients in tropical countries [1]. No melioidosis vaccine is currently available, and effectiveness of the recommendations for melioidosis prevention has not been evaluated to date [1]. In Thailand, evidence-based guidelines recommend that residents, rice farmers and visitors should wear protective gear such as rubber boots when in direct contact with soil and environmental water, and consume only boiled or bottled water [11]. Only a small proportion of people follow such recommendations, even though the Ministry of Public Health (MoPH) Thailand has consistently recommended them [11]. In a previous focus group study, we identified barriers to adopting recommended preventive behaviours [11]. The main barriers are categorized into five domains: (i) knowledge, (ii) beliefs about consequences, (iii) intention and goals, (iv) environmental context and resources, and (v) social influence [11]. People have little knowledge of melioidosis, believe that there is little or no harm in not adopting the recommended preventive behaviours, and are not inclined to use boots while working in muddy rice fields [11]. Using the Theoretical Domains Framework [12,13] and the Behaviour Change Wheel [14,15], we previously identified intervention options and modes of delivery, and developed a multifaceted prevention programme aimed at changing behaviour to prevent melioidosis, based on the local context in Thailand [16]. We also reported the protocol and feasibility of the programme in a pilot group of diabetics in northeast Thailand [17].

Here, we reported the outcomes of the PREMEL study, aiming to evaluate whether a multifaceted prevention programme in diabetics would reduce hospital admissions involving infectious diseases and culture-confirmed melioidosis infections. A cluster-randomized design was selected because the intervention was the behavioural support group session. A step-wedge design was selected because of the recommendation of the ethical committees to provide the intervention to all participants, and inability to achieve the target power and follow the recommendation of the ethical committees in a parallel design.

## Methods

### Ethics statement

All participants provided individual written consent before enrollment. The trial was approved by the Institute for the Development of Human Research Protections, Ministry of Public Health, Thailand (ref 189/2557) and Oxford Tropical Research Ethics Committee, University of Oxford, United Kingdom (ref 06–14). This trial is registered with ClinicalTrials.gov, number NCT02089152.

### Trial design and participants

From April 2014 to December 2018, we conducted a stepped-wedge cluster-randomized controlled behaviour change trial in Ubon Ratchathani province, northeast Thailand, where there was one provincial public hospital (Sunpasitthiprasong Hospital), three general public hospitals, 22 district hospitals and 317 Tambon Health Promoting Hospitals (THPHs). All hospitals also acted as primary care units (PCUs) that provide health promotion, prevention and medical treatment, including diabetic clinics, for communities. Clusters consisted of 116 PCUs in the province.

Diabetic patients aged from 18 to 65 years old presenting at diabetic clinics were invited to participate. Diabetes was defined as having fasting plasma glucose ≥126 mg/dl, HbA1c ≥6.5%, 2-hour plasma glucose (PG) ≥200 mg/dl during an oral glucose tolerance test, or classic

symptoms of hyperglycaemia with a random PG ≥200 mg/dl. Patients who had been diagnosed with melioidosis and had not completed oral-eradicative treatment for melioidosis were excluded.

## Randomisation

Diabetic patients were approached individually. Those consenting to enrollment in the study were asked for blood samples to test for HbA1c. We informed them that they would be randomly assigned to a group intervention lasting about 50–60 minutes once during the study period and that the aim was to prevent infectious diseases. Participants were not given the name of the target disease (melioidosis) or details of the intervention before the intervention. We completed the enrollment in November 2014, and allocated year 2015 as pre-intervention period. We randomly allocated the PCUs to receive the intervention starting in March 2016, 2017 and 2018 (defined as group 1, 2 and 3, respectively; Figs 1 and 2). After the completion of enrollment, an independent statistician generated the randomization code and assigned clusters to sequences.

## Procedures

The intervention was a multifaceted prevention programme for melioidosis developed using two behavioral frameworks: Theoretical Domains Framework [12,13] and the Behaviour Change Wheel [14,15]. Details of the intervention have been published [17]. The programme was a small-group intervention, in which 6 to 20 diabetic patients at a time attended a behavioural support group session conducted by the study team. Each session lasted about 50 to 60 minutes. The aim was to deliver the intervention from March to May of each year, prior to the start of rainy season and rice farming in June in northeast Thailand. Each diabetic patient was contacted by a member of the study team who gave them the date and the venue of their behavioural support group session. As diabetic patients came early in the morning for fasting blood glucose (FBG) testing, we delivered the behavioural support group after patients had breakfast (after FBG testing) and while they were waiting to see the doctors. When necessary, we delivered the group session after patients saw the doctors.

   The objective of the intervention was to increase the frequency of the two recommended preventive behaviours: wearing boots while working in rice fields and drinking boiled or bottled water. The multifaceted prevention programme included 13 behaviour change techniques identified by a focus group study conducted in 2012 [16,17]. The behaviour change techniques include information about health consequences (e.g. explaining that not wearing boots while working in rice fields and that drinking untreated water can lead to an often fatal infectious disease called melioidosis), credible source (e.g. a high status professional in the government giving a speech that emphasises the importance of melioidosis prevention), adding objects to the environment (e.g. providing baby powder and long socks to alleviate the problem of discomfort due to heat and humidity when wearing boots), reconstructing the physical environment, instruction on how to perform a behaviour, demonstration of the behaviour, commitment, prompts/cues, self-monitoring of behaviour, goal setting, feedback on behaviour, feedback on outcome(s) and social support.

   The intervention package included six short videos, three pamphlets, and a calendar with a space for participants' individual photographs and self-pledge. The materials are publicly available online (https://dx.doi.org/10.6084/m9.figshare.5734155) [17]. Each participant also received a pair of long socks and a bottle of baby powder (to reduce uncomfortable feelings while wearing boots) and a 2-litre plastic ice bucket (commonly used to store boiled water to drink while working in rice fields). In each behavioural support group, participants received

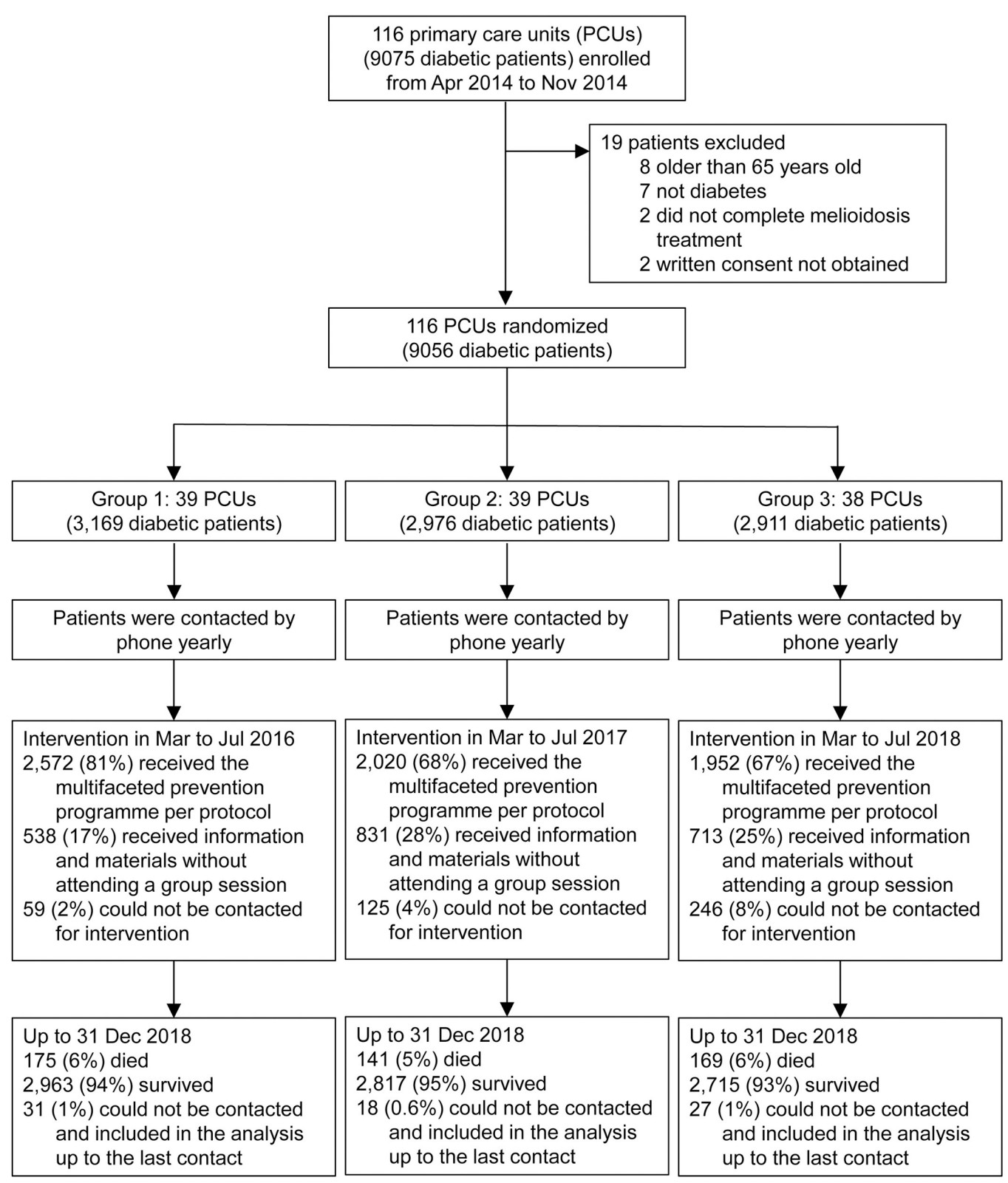

**Fig 1. Trial profile.**

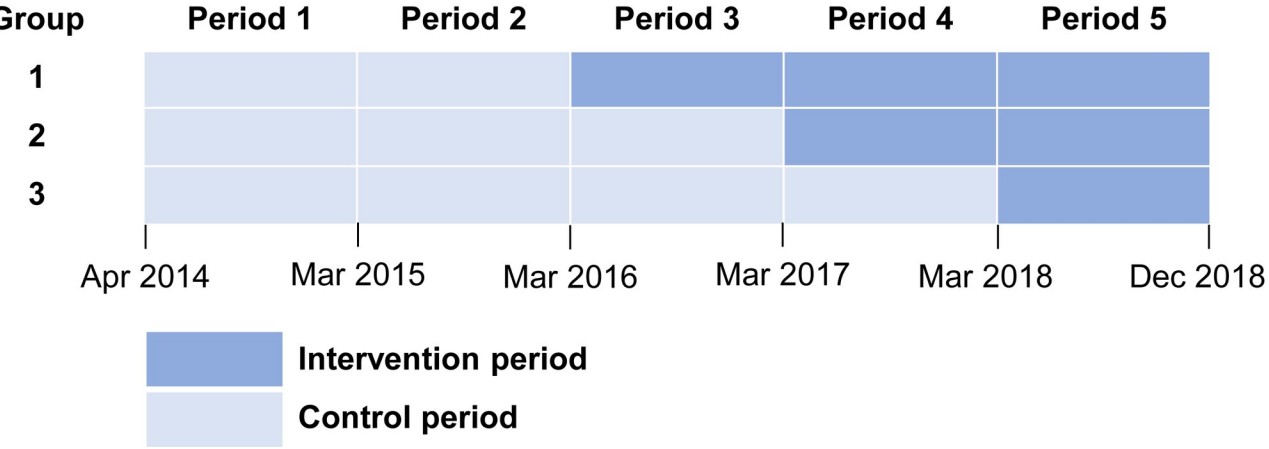

**Fig 2. Schematic of the PREMEL stepped-wedge cluster-randomised controlled trial.** PREMEL = multifaceted PREvention programme of MELioidosis in diabetics. No group received the intervention at baseline. Period 1 was the enrollment period, and period 2 was the baseline period. Clusters were randomly assigned to three groups that crossover to receive the intervention in March 2016, 2017 and 2018. Participants (diabetic patients enrolled) in each cluster group received a multi-faceted prevention programme once from March to July 2016, 2017 and 2018, respectively.

an introduction by a moderator, watched a series of brief videos, and had short group discussions at the end of each video. Participants then had a session in which they tried out multiple kinds of boots to identify the ones which would be most comfortable for wearing. Next, the study team took a photograph of each participant while wearing their boots and holding a kettle and gave participants their printed photographs. Finally, participants made their own calendar to act as a reminder tool for the recommended preventive behaviours. We asked participants to attach their individual photograph to the calendar and write their own pledge on the calendar themselves. Participants were asked to hang their calendar in their house. The moderator also stimulated group discussion before and after as well as during the sessions. Additionally, we provided social support by giving information to nurses, doctors, participants' relatives and village health volunteers in each participating PCU about the intervention and its potential benefits. We asked them to encourage participants to continue with the recommended behaviours.

### Outcomes

Two primary outcomes were hospital admissions involving infectious diseases and culture-confirmed melioidosis. Hospital admissions involving infectious diseases were determined using the International Statistical Classification of Diseases and Related Health Problems, 10th Revision, Thailand Modification (ICD-10-TM) [18] or final diagnoses made by attending physicians including the terms fever, febrile, infected, infection, abscess, pus, diarrhea, sepsis, pneumonia, cellulitis or diabetic foot. Secondary outcomes were all-cause mortality, and overall melioidosis (culture-confirmed melioidosis plus clinical melioidosis defined by attending physicians). We excluded continuing treatment from previous admissions, hospital-acquired infections and healthcare-associated infections by not including admissions occurring within 30 days of the discharge date of previous admissions. Culture-confirmed melioidosis was defined as the symptoms and signs of infection in association with at least one culture from any site positive for *B. pseudomallei*.

All diabetic patients were contacted by phone yearly. For diabetic patients who were admitted to hospitals, hospital admission data together with microbiology laboratory results were

obtained from hospitals in the province. To determine outcomes of participants up to 31 December 2018, we conducted the last phone contact on every participant from 1 January to 30 June 2019. To investigate whether deaths were associated with infectious diseases, we used ICD-10-DM, microbiology laboratory results and final diagnoses made by attending physicians for admissions occurring within 30 days prior to death. We also used cause of death informed by relatives via phone contact.

## Statistical analysis

The PREMEL study power calculation was performed with the assumption that, among diabetes, incidence rates of hospital admissions involving infectious diseases were 50 per 1,000 persons per year [19,20] and incidence rates of culture-confirmed melioidosis were 240 per 100,000 persons per year [10]. Using an alpha error of 0.05, a power of 0.8 and an intra-cluster correlation coefficient of 0.15, we calculated that we needed at least 9,000 diabetics from 30 diabetic clinics (300 diabetics per clinic). This design and sample size gave us 85% power to show a 30% reduction in incidence rates of hospital admissions involving infectious diseases and a 35% reduction in incidence rates of culture-confirmed melioidosis. During the enrollment period, we found that the number of diabetic patients we could enroll at each PCU was lower than we expected. Therefore, during the enrollment period, we adjusted and enrolled diabetic patients from 116 PCUs in the province.

We used both intention-to-treat (ITT) and per-protocol analyses. For the primary outcome of hospital admissions involving infectious diseases, we used multilevel mixed-effects negative binomial regression models adjusted for calendar time, and with a random effect for PCU and a random effect for repeated measures on the same diabetic patients [21,22]. This approach was recommended by Hemming et al [21,22] to allow for correlations between individuals in the same cluster and the dependence between individual measurements over the course of the study. In the ITT analysis, 1st March was set as the time of the intervention for the randomized clusters of those years. Patients were considered at risk from the enrollment until 31 Dec 2018. In the per-protocol analysis, participants who received a behavioural support group session for melioidosis prevention were defined as having received the intervention.

For the melioidosis and mortality outcomes we used multilevel mixed-effects Poisson regression models, adjusted for calendar time, with a random effect for PCU. This was because the multilevel mixed-effects negative binomial model did not converge [21,22]. In all models we performed interaction tests to evaluate whether treatment effects change over time [21,22].

We ran sensitivity analyses to evaluate whether the effectiveness of the intervention changed after adjusting for gender, age, diabetes duration and HbA1c level at enrollment using a multivariable regression model. We also evaluated whether the effectiveness would be observed when infections that are not plausibly related such as urinary tract infections were excluded. All analyses were performed using STATA version 14.2 (StataCorp, College Station, TX).

## Patient and public involvement

No patients or members of the public were involved in the design or conduct of our research. However, Ubon Ratchathani Provincial Public Health Office contributed to the planning stages. Our research uptake strategy included widespread engagement with key stakeholder groups; including Diabetes Association of Thailand and other melioidosis-endemic countries.

## Results

From April 2014 to November 2014, we enrolled 9,075 diabetic patients in 116 PCUs, representing the clusters, into the study (Fig 1).

**Table 1. Baseline characteristics at enrollment.**

| Characteristics | Group 1 | Group 2 | Group 3 | Total |
|---|---|---|---|---|
| Total no. of clusters | 39 | 39 | 38 | 116 |
| Total no. of participants | 3169 | 2976 | 2911 | 9056 |
| Sex, female | 2285 (72%) | 2194 (74%) | 2158 (74%) | 6637 (73%) |
| Age | | | | |
| 18 - <50 years | 872 (28%) | 829 (28%) | 778 (27%) | 2479 (27%) |
| 50 - <60 years | 1453 (46%) | 1308 (44%) | 1311 (45%) | 4072 (45%) |
| 60–65 years | 844 (27%) | 839 (28%) | 822 (28%) | 2505 (28%) |
| Known diabetes duration | | | | |
| <5 years | 1292 (41%) | 1235 (41%) | 1168 (40%) | 3695 (41%) |
| 5 - <10 years | 956 (30%) | 879 (30%) | 844 (29%) | 2679 (29%) |
| ≥10 years | 921 (29%) | 862 (29%) | 899 (31%) | 2682 (30%) |
| Diabetic control | | | | |
| No medication | 115 (4%) | 115 (4%) | 113 (4%) | 343 (4%) |
| Only oral medication | 2503 (79%) | 2369 (80%) | 2264 (78%) | 7136 (79%) |
| Insulin therapy | 551 (17%) | 492 (17%) | 534 (18%) | 1577 (17%) |
| HbA1C level | | | | |
| <7.0% | 706 (22%) | 698 (23%) | 664 (23%) | 2068 (23%) |
| 7.0–8.0% | 898 (28%) | 800 (27%) | 819 (28%) | 2517 (28%) |
| >8.0–9.0% | 603 (19%) | 565 (19%) | 565 (19%) | 1733 (19%) |
| >9.0% | 962 (30%) | 913 (31%) | 863 (30%) | 2738 (30%) |
| History of co-morbidities | | | | |
| Hypertension | 1999 (63%) | 1920 (65%) | 1728 (59%) | 5647 (62%) |
| Dyslipidemia | 1800 (57%) | 1665 (56%) | 1702 (58%) | 5167 (57%) |
| Kidney diseases | 176 (6%) | 152 (5%) | 235 (8%) | 563 (6%) |

Baseline characteristics of the 9,056 diabetic patients included in the study are summarised in Table 1, stratified by group. Their baseline characteristics were similar between groups. Overall, 73% were female, the median age was 55 years (IQR 49–60), 41% had known diabetes duration less than 5 years, 79% were taking only oral medication for their diabetes and 30% had poor diabetic control (HbA1c >9.0%) on enrollment. Fifty-three patients had history of culture-confirmed melioidosis and completed oral eradicative treatment prior to the enrollment.

A total of 39, 39 and 38 PCUs were randomized to groups 1, 2 and 3, in which the intervention was given from March to July in 2016, 2017 and 2018, respectively (Fig 2). Of 9,056 diabetic patients, 6,544 (72%) received a behavioural support group session. The study team delivered a total of 522 sessions, of which 177 (34%) had fewer than six participants per session, 237 (45%) had 6 to 20 participants, and 108 (21%) had more than 20 participants. Of 522 sessions, 408 (78%) were conducted from March to May, 94 (18%) in June, and 20 (4%) in July. Participants who received a behavioural support group session were older and had a higher proportion of female, lower proportion of insulin therapy (16% vs 22%) and lower proportion of poor diabetic control (29% vs 33%) compared with those who did not receive a behavioural support group session (Table 2).

Of 2,512 diabetic patients who did not receive a behavioural support group session, 1,696 (68%) did not meet the study team, but received details of the behavioural support group by phone and received materials by delivery via village health volunteers. Another 386 (15%) diabetic patietns met the study team, received the materials and details of the behavioural support

**Table 2. Characteristics of participants who received a behavioural support group session for melioidosis prevention.**

| Characteristics | Received the intervention (n = 6544) | Did not receive the intervention (n = 2512) | P value |
|---|---|---|---|
| Groups | | | |
| Group 1 | 2572 (39%) | 597 (24%) | <0.001 |
| Group 2 | 2020 (31%) | 956 (38%) | |
| Group 3 | 1952 (30%) | 959 (38%) | |
| Sex, female | 4998 (76%) | 1639 (65%) | <0.001 |
| Age | | | |
| 18 - <40 years | 1713 (26%) | 766 (30%) | <0.001 |
| 40 - <55 years | 2972 (45%) | 1100 (44%) | |
| 55–65 years | 1859 (28%) | 646 (26%) | |
| Diabetes duration | | | |
| <5 years | 2645 (40%) | 1050 (42%) | 0.14 |
| 5 - <10 years | 1974 (30%) | 705 (28%) | |
| ≥10 years | 1925 (29%) | 757 (30%) | |
| Diabetic control | | | |
| No medication | 248 (4%) | 95 (4%) | <0.001 |
| Only oral medication | 5263 (80%) | 1873 (75%) | |
| Insulin therapy | 1033 (16%) | 544 (22%) | |
| HbA$_{1c}$ level | | | |
| <7.0% | 1523 (23%) | 545 (22%) | <0.001 |
| 7.0–8.0% | 1841 (28%) | 676 (27%) | |
| >8.0–9.0% | 1283 (20%) | 450 (18%) | |
| >9.0% | 1897 (29%) | 841 (33%) | |
| History of co-morbidities | | | |
| Hypertension | 4125 (63%) | 1522 (61%) | 0.03 |
| Dyslipidemia | 3767 (58%) | 1400 (56%) | 0.10 |
| Kidney diseases | 362 (6%) | 201 (8%) | <0.001 |

Data are n (%) or median (interquartile range).

group personally, but declined to attend a behavioural support group session. The most frequent reasons for declining were that they did not want to wait for the session (while a session was already running), feared that they would miss their place in the queue for the doctor, wanted to go back home immediately (for cases that had already seen a doctor, and could attend a session after that), and unknown reasons. The other 250 (10%) diabetic patients died prior to the intervention period and 180 (17%) diabetic patients could not be contacted.

As of the end of December 2018, 8,495 (94%) patients survived, 485 (5%) died, and 76 (1%) could not be contacted. Of 76 patients who could not be contacted, six (0.1%) withdrew consent during the study period and data up to the last follow-up were used in the analyses. Total duration of follow-up period was 38,457 person-years.

For the primary outcome, 3,335 hospital admissions involving infectious diseases occurred in 2,195 patients. The most common diagnoses were acute gastroenteritis (582 admissions), pneumonia (367 admissions), post-traumatic wound infection (302 admissions), urinary tract infection (287 admissions), cellulitis (251 admissions) and unspecified fever (241 admissions; S1 Table). The rate of hospital admission involving infectious diseases was 87 (95%CI 84–90) admissions per 1,000 person-years. In the ITT analysis, the intervention was not associated with the incidence rate of hospital admissions involving infectious diseases (p = 0.79; Table 3). In the per-protocol analysis, diabetic patients who received a behavioural support group

**Table 3. Outcomes of the study.**

|  | Adjusted (for time) incidence rate ratio (95% CI) | Adjusted (for time and other risk factors*) incidence rate ratio (95% CI) |
|---|---|---|
| **Intention-to-treat analysis** |  |  |
| Primary outcomes |  |  |
| Hospital admissions involving infectious diseases | 0.98 (0.87–1.11) | 0.98 (0.87–1.10) |
| Culture-confirmed melioidosis | 0.65 (0.29–1.47) | 0.66 (0.29–1.50) |
| Secondary outcomes |  |  |
| Overall melioidosis | 0.73 (0.37–1.44) | 0.74 (0.38–1.45) |
| Mortality | 0.97 (0.74–1.28) | 0.98 (0.74–1.29) |
| **Per-protocol analysis** ** |  |  |
| Primary outcomes |  |  |
| Hospital admissions involving infectious diseases | 0.89 (0.80–0.99) | 0.90 (0.81–1.00) |
| Culture-confirmed melioidosis | 0.85 (0.41–1.76) | 0.96 (0.46–1.99) |
| Secondary outcomes |  |  |
| Overall melioidosis | 0.57 (0.31–1.08) | 0.65 (0.35–1.22) |
| Mortality | 0.54 (0.43–0.68) | 0.56 (0.44–0.71) |

CI = confidence interval

* Adjusted for age, sex, known diabetes duration and HbA1c level

** Participants who received a behavioural support group session for melioidosis prevention were defined as received the intervention per protocol.

session had 11% lower incidence rate of hospital admissions involving infectious diseases (incidence rate ratio [IRR] 0.89; 95%CI 0.80–0.99, p = 0.03). The primary outcome of culture-confirmed melioidosis occurred in 58 patients. Fifty-seven patients admitted to hospitals and one patient had localized melioidosis infection of their hand with pus culture positive for *B. pseudomallei*. The patient was treated successfully as an outpatient case with cotrimoxazole-sulphamethoxazole as the oral eradicative treatment. In both ITT and per-protocol analyses, the intervention was not associated with the incidence of culture-confirmed melioidosis (p = 0.30 and p = 0.66, respectively).

The secondary outcome of all-cause mortality occurred in 485 patients. 15 of 58 (26%) patients with culture-confirmed melioidosis and none of 22 (0%) patients with clinical melioidosis died within 30 days of the hospital admissions of melioidosis. Of 485 deaths, 213 (44%) occurred within the hospitals, 198 (41%) occurred within 30 days after the last hospital admission and 74 (15%) occurred at home without hospital admission within 30 days prior to death. Of 198 who died within 30 days after the last hospital admission, 101 (51%) died on the hospital discharge date, 49 (25%) died within 1 to 7 calendar days after the hospital discharge, and 48 (24%) died within 8 to 30 calendar days after the hospital discharge. We found that 217 (45%) of 485 deaths were possibly related to infectious diseases, including septic shock recorded by attending physicians in 94 deaths (19%) (S2 Table). In the ITT analysis, implementation of the intervention was not associated with mortality (p = 0.85). In the per-protocol analysis, patients who received the multifaceted prevention programme had 46% lower rate of mortality (IRR 0.54; 95%CI 0.43–0.68, p<0.001). As we adjusted for calendar time, we observed that the rate of hospital admissions involving infectious diseases and mortality rose over time.

The secondary outcome of overall melioidosis occurred in 80 patients (58 culture-confirmed melioidosis and 22 clinical melioidosis). Of 53 patients who had history of culture-confirmed melioidosis and completed oral eradicative treatment prior to the enrollment, none

had culture-confirmed melioidosis and three had clinical melioidosis during the study period. In both ITT and per-protocol analysis, implementation of the intervention was not significantly associated with the incidence rate of melioidosis (p = 0.37 and p = 0.09, respectively).

In a sensitivity analysis, using per-protocol analysis and multivariable regression models, we found that male gender, older age, longer known diabetes duration and higher HbA1c levels on enrollment were associated with incidence of hospital admissions involving infectious diseases (S3 Table) and mortality (S4 Table). Male gender, longer known diabetes duration and higher HbA1c level on enrollment were also associated with overall melioidosis (S5 Table). We also conducted a pre-specified analysis excluding hospital admissions involving infections that are not plausibly related to the intervention such as urinary tract infections, and similar results were observed (S6 Table).

## Discussion

This is the first trial of a behavioural intervention to prevent melioidosis to the authors' knowledge. Clear benefits of the multifaceted prevention programme for melioidosis could not be observed. It shows that diabetic patients who receive a behavioural support group session for melioidosis prevention have lower rates of hospital admissions related to infectious diseases and of all-cause mortality. Rates of culture-confirmed melioidosis were not statistically different. Our study also did not observe an intervention effect in the ITT analyses.

The absence of a clear intervention effect could be due to the type of the intervention and lack of statistical power. This is probably because a proportion of enrolled patients did not participate in a behavioural support group session, and only the patients who received a behavioural support group session adopted the recommended behaviours in significant numbers [16]. Providing information and materials without attending a behavioural supportive group probably only had a minimal effect on behaviour [14,15]. The study had a high proportion of female diabetic patients, while male diabetic patients are at a higher risk of melioidosis [1–3]. The reason for this could be that we met less male diabetic patients as males are more likely than females to miss appointments at diabetic clinics [23,24]. In addition, based on our experience, males are more likely to decline invitations to participate in studies than females. A high proportion of female diabetic patients in prospective studies (ranging from 58% to 75%) is also observed in other prospective studies in diabetics in Thailand [25]. Nonetheless, both male and female diabetic patients are primary targets for interventions directed at melioidosis prevention. It is also possible that a single behavioural support group session and component behavior change techniques used were not adequate. This suggest that the invitation for the intervention and prevention programme may need to be more proactive and persuasive for male diabetic patients. Modification or addition of the behaviour change techniques used, and more frequent intervention may also be needed.

Our findings suggest that wearing protective gear and drinking boiled water could have a wide impact on infectious diseases and overall health; however, the observed size of effect (e.g. IRR 0.54 for all-cause mortality) could also be attributed to other confounding factors. For example, diabetic patients may also improve their glycemic control and diabetes self-management [26], on top of wearing boots and drinking boiled water, after receiving a behavioural support group session. This may result from hearing information about the consequence of melioidosis and its association with diabetes and poor diabetic control from credible sources, and stories from relatives of fatal cases in the videos which were part of the behavioural support group session. It is also possible that diabetic patients who received a behavioural support group session might have better self-care behaviour or expose themselves less often to risky environments than those who did not receive the intervention.

We show that poor diabetic control (HbA1c >9.0%) and longer known diabetic duration at enrollment are significantly associated with hospital admissions related to infectious diseases, overall melioidosis and all-cause mortality. The association between poor diabetic control and all-cause mortality is consistent with previously published studies [27]. Infection-related deaths as a proportion of all-cause mortality estimated in our study (45%) was much higher than previous reports, ranging from 4 to 15% from the cohort in the U.S. [28] and 16% in the U.K. [29]. It is likely that the proportion of infection-related deaths in diabetes is higher in low and middle-income countries, where the burden of sepsis is estimated to be highest [30]. The difference could also be associated with the difference in training for International Statistical Classification of Diseases coders and limited data on cause of death in Thailand.

The proportion of diabetic patients with poor diabetic control in our study (30%) is relatively higher than the previous reports in Thailand [25,31] This could be because we tested HbA1c in 100% of the diabetic patients on enrollment. Previous studies in Thailand found that only 50% to 78% of diabetic patients had HbA1c tested and diabetic patients with repeatedly high fasting blood glucose may not be tested for HbA1c levels [24,25] The findings of increasing rate of hospital admission involving infectious diseases and mortality as the study period progressed (i.e. a rising tide phenomenon [21,22]) is consistent with our expectation that those events would occur more often as diabetic patients age and have a longer duration of living with diabetes.

This study has several strengths: the large sample size; the large number of PCUs; and the use of two behavioral frameworks: Theoretical Domains Framework [12,13] and the Behaviour Change Wheel [14,15] for the development of the intervention [16,17]. Strong support from PCUs and village healthcare volunteers, and the high rate of follow-up via phone contact, increased the reliability of hospital admissions and mortality endpoints over the study period. Limitations include the impossibility of blinding the intervention after unmasking, and the inability to record the data of non-responders to an invitation to the study, to repeat HbA1c, to have more than one behavioural support group session, to have more frequent follow-up, and to measure recommended behaviours. A high number of enrolled patients later declined to receive or wait around for the intervention. They were more likely to be male and have poor diabetic control. We also found that these participants were more likely to have hospital admissions involving infectious diseases, melioidosis and fatal outcome. Due to pragmatic reasons, we could not ensure the number of participants per session were always between six to 20. One of the major weakness is the lack of monitoring and reminder with only a yearly phone call. Lost opportunity to reinforce at regular clinic visits for diabetes which are more frequent than yearly may have improved the outcomes. Had we had more human resources, time and budget, we would have included more phone calls and reminder text messages including during the rainy season. The study team had eight research assistants, and the implementation of the intervention, conducting yearly phone calls, and following data on hospital admissions of 9,000 participants already required the full capacity of the study team. We also provided calendars as a reminder so that they could note their activities on a daily basis. ICD coding can also be diagnostically inaccurate because of the subjective nature of ICD certification practices and limited training. The association between receiving a behavioural support group session for melioidosis prevention and lower rates of hospital admissions related to infectious diseases was not strong, and the impact of the intervention on hospital admission related to infectious diseases should be considered carefully.

In conclusion, clear benefits of this multifaceted prevention programme for melioidosis were not observed. Successful and cost-effective interventions are still needed though challenging to design and implement. We propose that more compelling invitations for the intervention, modification of or addition to the behaviour change techniques used, and more frequent

intervention may be needed. In addition, alternative interventions including targeted use of antimicrobial prophylaxis [32], and more general clinical and public health interventions such as better control of diabetes and improvement of water treatment [33] should also be considered.

## Supporting information

**S1 Table. ICD-10-TM codes and criteria used to determine that hospitals admissions were possibly related to infectious diseases.**
(DOCX)

**S2 Table. ICD-10-TM codes and criteria used to determine that deaths were possibly related to infectious diseases.**
(DOCX)

**S3 Table. Factors associated with hospital admissions involving infectious diseases.**
(DOCX)

**S4 Table. Factors associated with mortality.**
(DOCX)

**S5 Table. Factors associated with overall melioidosis.**
(DOCX)

**S6 Table. Outcomes of the study excluding infections that are not plausibly related to the intervention.**
(DOCX)

## Acknowledgments

We thank all patients, their relatives, staff of all PCUs who participated in the study, and the Independent Trial Steering Committee, including Prof Ploenchan Chetchotisakd from Khon Kaen University, Thailand; Prof Sarah Walker from University College of London, United Kingdom; Dr Rungrueng Kitphati and Dr Saowapak Hinjoy from Ministry of Public Health, Thailand; Alisa Suphan, Yaowaluck Phokalataweepong and Supaporn Paerwong from Public Health Office, Ubon Ratchathani, Thailand. We thank our study team (Wilasinee Thongklang, Thidarat Phosri, Meree Chuenjit, Jirawat Narerat, Santipong Gropthong, Jiraporn Mungmai, Chompunoot Suwannasala, Boodsayakorn Malai, Mayura Malasit, Rungthiwa Chaken, Supaporn Saennam, Putchara Kontong, Piyapong Sangsai, Sudarat Galaboot, Nusara Patichai and Jirawan Bunthow) the laboratory team (Areeya Faosap, Yaowaret Dokket, Sukhumal Pewla-orng, Malinee Oyuchua and Premjit Amornchai) and the operation team (Jintana Suwanna-pruek, Papachaya Phuangsombat and Maliwan Hongsuwan) of Mahidol-Oxford Tropical Medicine Research Unit at Ubon Ratchathani, Thailand. We also thank Viriya Hantrakun, Noppawan Lumjarungsimalert, Pasathorn Sirithiranont, Thatsanun Ngernseng, Prapass Wannapinij, Cherry Lim, Apiraya Ammarapal, Nantawat Sriwattana and Chutporn Tutsanawiwat for their technical support.

## Author Contributions

**Conceptualization:** Direk Limmathurotsakul.

**Data curation:** Pornpan Suntornsut, Gumphol Wongsuvan.

**Formal analysis:** Direk Limmathurotsakul.

**Funding acquisition:** Direk Limmathurotsakul.

**Investigation:** Gumphol Wongsuvan.

**Methodology:** Pornpan Suntornsut, Prapit Teparrukkul, Wipada Chaowagul, Susan Michie, Nicholas P. J. Day, Direk Limmathurotsakul.

**Project administration:** Pornpan Suntornsut, Prapit Teparrukkul, Wipada Chaowagul.

**Supervision:** Susan Michie, Nicholas P. J. Day, Direk Limmathurotsakul.

**Writing – original draft:** Pornpan Suntornsut, Direk Limmathurotsakul.

**Writing – review & editing:** Pornpan Suntornsut, Prapit Teparrukkul, Gumphol Wongsuvan, Wipada Chaowagul, Susan Michie, Nicholas P. J. Day, Direk Limmathurotsakul.

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
