## [Decision Letter · Decision Letter 0]

16 Feb 2021

Dear Dr. Limmathurotsakul,

Thank you very much for submitting your manuscript "Effectiveness of a multifaceted prevention programme for melioidosis in diabetics (PREMEL): a stepped-wedge cluster-randomised controlled trial" for consideration at PLOS Neglected Tropical Diseases. As with all papers reviewed by the journal, your manuscript was reviewed by members of the editorial board and by several independent reviewers. In light of the reviews (below this email), we would like to invite the resubmission of a significantly-revised version that takes into account the reviewers' comments. 

The reviewers recommend substantial changes to the manuscript, including the conclusion, recommending caution in not over-stating the potential benefits of the intervention, and justification for inclusion of melioidosis cases that were not microbiologically confirmed. Despite the negative findings for the study, these are potentially important findings to share after addressing the reviewers’ comments. The authors could discuss why the interventions did not result in significant differences, with respect to study design or biological factors.

We cannot make any decision about publication until we have seen the revised manuscript and your response to the reviewers' comments. Your revised manuscript is also likely to be sent to reviewers for further evaluation.

Sincerely,

Yunn-Hwen Gan

Associate Editor

Anna Ralph

Deputy Editor

Note that reviewer 2 has consolidated his/her comments at a single section at the end instead of breaking into each manuscript section.

Reviewer's Responses to Questions

**Key Review Criteria Required for Acceptance?**

**Methods**

-Are the objectives of the study clearly articulated with a clear testable hypothesis stated?

-Is the study design appropriate to address the stated objectives?

-Is the population clearly described and appropriate for the hypothesis being tested?

-Is the sample size sufficient to ensure adequate power to address the hypothesis being tested?

-Were correct statistical analysis used to support conclusions?

-Are there concerns about ethical or regulatory requirements being met?

Reviewer #1: I am satisfied with the methods that have been employed.

The statistical analysis is very complicated and I lack the expertise to offer a sensible comment on the statistical methods that have been employed. I hope that one of the other reviewers have more expertise so that this might be critically assessed.

Reviewer #2: All good

**Results**

-Does the analysis presented match the analysis plan?

-Are the results clearly and completely presented?

-Are the figures (Tables, Images) of sufficient quality for clarity?

Reviewer #1: The results are presented in a detailed manner and in general, I am happy with them.

Although I might suggest that the authors try not to duplicate results in the text and the tables/figures. For instance, lines 278-281 is repeated - almost word for word - in figure 1 and doesn't add much to the study's interpretation (0.2% of the cohort were excluded). There are several other examples where data are duplicated in text and in a table or figure like this.

Reviewer #2: Yes

**Conclusions**

-Are the conclusions supported by the data presented?

-Are the limitations of analysis clearly described?

-Do the authors discuss how these data can be helpful to advance our understanding of the topic under study?

-Is public health relevance addressed?

Reviewer #1: This is my major concern with the paper.

I think the authors have performed an impressive study, but they have over-egged their - in my opinion - negative results. To me the studied intervention - if significant at all - appears to be a very low-value intervention.

The primary aim of the study was to evaluate the effectiveness of a prevention programme for melioidosis.

22/80 (27.5%) of the patients who were said to have melioidosis had a "clinical diagnosis".

As the authors themselves note in lines 94-95, a positive culture for B. pseudomallei is the gold standard for diagnosis.

So why then do they include the "clinical diagnosis" of melioidosis cases in their analysis? I would be quite confident that a significant proportion of the patients with "clinical melioidosis" did not actually have melioidosis. How then to explain that their case-fatality rate was 0 (compared to 26% among the culture-confirmed group). Even in the very well-resourced Australian health system the case-fatality rate of melioidosis is about 10% of those admitted. I would suggest that the reason that the case-fatality rate in the patients with "clinical melioidosis" is so low is that many did not actually have melioidosis. Indeed, the authors state that the reason that you would want to prevent melioidosis is that it is often fatal; in lines 86-88 they cite a modelling paper that suggests that 54% of cases of melioidosis die. 

The patients with "clinical melioidosis" in this cohort actually did pretty well! If this is the case, perhaps there is less need to worry about preventing it?

Even including the clinically diagnosed cases in a per-protocol analysis in a sample of 9075 patients the IRR is 0.57 (95%CI 0.31-1.08), p=0.09. This is not, as the authors state, "borderline evidence of a lower incidence rate" it is statistically not significant. Even if the authors suggest that this "trend" is a type 2 error, the size of the sample that you would need to prove that the effect was statistically significant suggests that the actual impact of the intervention is quite limited. If you can’t show it in 38,457 person-years of follow-up, any putative benefit much be tiny. What is the number needed to treat (NNT)?

The two main interventions that the authors were keen to implement (drinking bottled/boiled water and wearing boots - lines 172-173) were said to reduce hospital admissions for infectious diseases (IRR 0.89 (95%CI 0.80-0.99, p=0.03), but how much was this simply determined by gastroenteritis admissions, (the most common cause for admission with infection) which could just be explained by getting the participants to drink boiled/bottled water. No need to fuss around with all the education about wearing boots. Additionally, how do the authors propose the wearing of boots and drinking bottled/boiled water reduce urinary tract infections?

The patients who did not receive the intervention were more likely to be male and more likely to have poorer glycaemic control (among other significant differences). Although the authors have used complicated statistical methods to control for the contribution of these factors, lacking the statistical knowledge to critically appraise this, I would want to be confident that the patients who declined to receive or wait around for the intervention are not just the patients who are going to make poorer health decisions anyway. Male gender is a recognised risk factor for poor health outcomes and poor glycaemic control may also be a proxy maker for poor access to care/adherence. Indeed, the authors do acknowledge this as a potential limitation in the study (lines 400-402). I think this discussion could be expanded. 

Finally, I am not convinced by their health economic argument. The statistically non-significant benefit of the intervention would – the authors “estimate” – be cost-effective if policy makers were willing to pay $7000 per QALY. If this funding were directed elsewhere – to measures that improve glycaemic control for instance (lifestyle and pharmacological interventions) or prevent the development of diabetes, not only would this reduce the incidence of melioidosis but also the cardiovascular deaths and other complications of metabolic syndrome which have far more impact at a community level than melioidosis (56 confirmed cases in 38,457 years of patient follow up)

Reviewer #2: Yes

**Editorial and Data Presentation Modifications?**

Reviewer #1: (No Response)

Reviewer #2: (No Response)

**Summary and General Comments**

Reviewer #1: I would like to congratulate the authors on an ambitious and impressive piece of work.

However, I am not at all convinced by the article's conclusions as they presently stand.

Even with 38,457 person-years of follow-up, using a per-protocol analysis and included "clinical" cases of melioidosis they have not been able to show that the intervention significantly reduced the incidence of melioidosis. 

There was a reduction in infections requiring hospitalisation, but what proportion of the reduction in infectious diseases admissions can be explained by the bottled water intervention alone on the most common infection, gastroenteritis?

Improving access to clean water may be a simpler and cost-effective intervention at a population level.

The paper certainly merits publication and the melioidosis research community will find the work very interesting. However, I think revision of their conclusions, downplaying the putative success of the intervention significantly would make the paper more credible.

Reviewer #2: The authors conducted a cluster randomised trial using step wedge design to study the impact of a behavioural intervention on hospitalisation due to infectious diseases and incidence of melioidosis among diabetic patients in Northern Thailand.

The authors acknowledged the imbalance in gender, and larger group size than predefined that may have influenced learning. 

A few comments:

(1) Can the authors explain why recruitment ended in 2014 but randomisation only occurred in 2016?

(2) Did the authors exclude all diabetic patients with known melioidosis? Exclusion criteria stated that those yet to complete antibiotic were excluded. If they did not, what proportion with known previous melioidosis was enrolled? Even with completion of coytrimoxazole, there is still a 5% relapse rate, which may have affected the incidence of melioidosis.

(3) Can the authors clarify why they did not conduct the intervention year round which could lead to better recruitment and better power as even if rainy season has started, behaviour change could still reduce infections? In fact, it may be more effective as the potential outcomes may be more real.

(4) One of the major weakness is the lack of monitoring and reminder with only a yearly phone call. Lost opportunity to reinforce at regular clinic visits for diabetes which are more frequent than yearly may have improved the outcomes. Can the authors explain why they decided only on yearly phone call? Why did they not ask for diary; reminder text messages; more frequent phone calls e.g. 3 monthly; reminder during rainy season?

(5) Can the authors explain why they thought obtaining cause of death from relatives may contribute to reliable data on secondary outcome of mortality? What was the average educational background of the family members?

(6) Primary outcome included infections like urinary tract infections that are not plausibly related to wearing boots and drinking boiled or bottled water. These are also not in pre-defined primary outcome. Can the authors explain the basis of inclusion? Can they do sensitivity analysis by excluding infections that are not plausibly related?

(7) There is no effect on melioidosis. I think the authors should accept that and not claim borderline or trend to siginficance in any part of the manuscript including abstract.

PLOS authors have the option to publish the peer review history of their article (what does this mean?). If published, this will include your full peer review and any attached files.

Reviewer #1: No

Reviewer #2: No
---

## [Decision Letter · Decision Letter 1]

29 Apr 2021

Dear Dr. Limmathurotsakul,

Thank you very much for submitting your revised manuscript "Effectiveness of a multifaceted prevention programme for melioidosis in diabetics (PREMEL): a stepped-wedge cluster-randomised controlled trial" for consideration at PLOS Neglected Tropical Diseases. As with all papers reviewed by the journal, your manuscript was reviewed by members of the editorial board and by several independent reviewers. The reviewers appreciated the attention to an important topic. Based on the reviews, we are likely to accept this manuscript for publication, providing that you modify the manuscript according to the review recommendations. 

Dear authors, thank you for your major revision that has addressed the major issues pointed out by both reviewers. To further improve the clarity and message of the manuscript, please address the 3 points raised by reviewer 1, and if there is disagreement with the reviewer, please justify. The public message from this study remains clear: that successful, low-cost interventions are challenging in this community and the authors should spare a few more sentences to describe the socioeconomic challenges and perhaps new avenues of intervention, eg through better control of diabetes or targeted use of antimicrobial prophylaxis (e.g. see https://pubmed.ncbi.nlm.nih.gov/29340327/) or water treatment (e.g. https://www.ncbi.nlm.nih.gov/pmc/articles/PMC3741262/), that may be more effective. 

Another minor point: Please delete the last sentence "Further study may be required" from the abstract as it adds no value.

Please revise carefully and to the points raised as this will be the last revision.

Sincerely,

Yunn-Hwen Gan

Associate Editor

Anna Ralph

Deputy Editor

Dear authors, thank you for your major revision that has addressed the major issues pointed out by both reviewers. To further improve the clarity and message of the manuscript, please address the 3 points by reviewer 1, and if there is disagreement with the reviewer, please justify. The public message from this study remains clear: that intervention is not easy in this community and the authors should spare a few more sentences to describe the social economic challenges and perhaps new avenues of intervention, eg through better control of diabetes that may be more effective. 

Another minor point: Please delete the last sentence "Further study may be required" from the abstract as it adds no value.

Please revise carefully and to the points raised as this will be the last revision.

Reviewer's Responses to Questions

**Key Review Criteria Required for Acceptance?**

**Methods**

-Are the objectives of the study clearly articulated with a clear testable hypothesis stated?

-Is the study design appropriate to address the stated objectives?

-Is the population clearly described and appropriate for the hypothesis being tested?

-Is the sample size sufficient to ensure adequate power to address the hypothesis being tested?

-Were correct statistical analysis used to support conclusions?

-Are there concerns about ethical or regulatory requirements being met?

Reviewer #1: See summary and general comments.

**Results**

-Does the analysis presented match the analysis plan?

-Are the results clearly and completely presented?

-Are the figures (Tables, Images) of sufficient quality for clarity?

Reviewer #1: See summary and general comments.

**Conclusions**

-Are the conclusions supported by the data presented?

-Are the limitations of analysis clearly described?

-Do the authors discuss how these data can be helpful to advance our understanding of the topic under study?

-Is public health relevance addressed?

Reviewer #1: See summary and general comments.

**Editorial and Data Presentation Modifications?**

Reviewer #1: Minor revision. See summary and general comments.

**Summary and General Comments**

Reviewer #1: Thank you for the opportunity to review the revised manuscript by Limmathurotsakul et al. I appreciate the time that they have take to address the reviewers’ comments. I have only a few comments to make.

While they have pleasingly re-presented the study as a negative one, rather than a borderline positive one, I would suggest that they have not gone quite far enough, but I will leave it to the Editor to adjudicate!

1. In the abstract they say that in the per-protocol analysis the patients receiving the intervention had a lower incidence of hospital admissions involving infectious diseases (OR: 0.89, 95%CI 0.80-0.99, p=0.03). 

In supplementary table 6, they present an OR of 0.89, (95%CI 0.80-1.0) and do not present a p value (that I can easily see).

They also say that they have “Excluded certain infectious and parasitic diseases (A15-A23, A25-A48, A50-A99, B00-B64, B85-B97), eye infection (ICD-10-TM H10, H16, H44.0), Infective otitis externa (ICD-10-TM H60, H65, H66), endocarditis (ICD-10-TM I01, I33, I38, I39), acute upper respiratory infections (ICD-10-TM J00-J06), influenza, viral pneumonia, pneumonia due to Streptococcus pneumoniae, Haemophilus influenzae and other specified infectious organisms (ICD-10-TM J09-J14, J16), acute bronchitis and acute bronchiolitis (ICD-10-TM J20-J21), (acute) cholecystitis (ICD-10-TM K80, K81), urinary tract infection (ICD-10-TM N13.6, N15.1, N30, N39.0) and infection and inflammatory reaction due to other internal prosthetic devices, implants and grafts (ICD-10-TM T85.7)” that are not plausibly related to the intervention (boots and bottled water)

However, I think I am right to say that they do not seem to have excluded a variety of infections that it would be quite a stretch to say are prevented by boots and bottled water. These would appear to include bone and joint infections and fever “unspecified”, among others. 

It appears that they are just keen to show “some” effect of the intervention. 

My strong advice would be to delete this from the abstract; if they are determined to link their intervention to a reduction in infectious diseases admissions, please list the infections that they are saying HAVE been prevented by the boots and bottled water (which I would argue might be limited to skin and soft tissue infections of the feet and their complications and gastroenteritis). Then present the odds ratio for the reduction in the incidence of these diseases.

2. I really think they should delete all references to “clinical melioidosis” in the paper. I am cognisant of the challenges of obtaining a microbiological diagnosis in regional Thailand, but to the melioidosis audience (who will be the main readers of the paper), this just reduces the paper's credibility. Especially as the case-fatality rate in these 25 patients was 0%. Even in Australia’s well resourced health system with access to ICU and ECMO the case fatality rate of hospitalised patients is about 10%. This suggests that many - if not most - of these patients with “clinical melioidosis” don’t have melioidosis at all. The CFR in the confirmed cases in this cohort was 26%, which is similar to relevant studies from the literature. 

Alternatively, the authors could produce a very interesting case series on how they managed to reduce the CFR to 0% in these clinical melioidosis patients. We’d be interested to know!

3. The authors have had a really good go at providing an intervention that will reduce the incidence of melioidosis. They are to be commended for their efforts and the work is a valuable addition to the melioidosis literature.

However, fundamentally this is a negative study. If they cannot show a benefit of the intervention in the controlled study environment, how would prevention strategies fare in the Real-World setting?

About 30% of the patients in the cohort had poor glycaemic control, so they are struggling with their diabetes management. Meanwhile, 27% of enrolled patients did not have the time/understanding/interest to attend the education session.

Delivering interventions to this population is obviously going to be challenging, yet this does not seem to have been acknowledged, rather “Stronger invitations for the intervention, modification or addition of the behaviour change techniques used, and more frequent intervention may be needed”.

I do not want to seem nihilistic, but I think given that despite the dedicated work of the study team, they could not show a benefit. Therefore I would think at least a sentence or two about the cost-effectiveness of melioidosis prevention at a population level would be appropriate. My back of the envelope calculations suggest that culture-confirmed melioidosis occurred in this cohort at a rate of 151/100,000/year. Would finite health Dollars/Baht be better spent on other public health strategies that address the social determinants of health, health literacy and access to care that affect not only melioidosis but other communicable and non-communicable diseases?

PLOS authors have the option to publish the peer review history of their article (what does this mean?). If published, this will include your full peer review and any attached files.

Reviewer #1: No

Figure Files:

Data Requirements:

Reproducibility:

References

---

## [Editor Report · Decision Letter 2]

4 Jun 2021

Dear Dr. Limmathurotsakul,

We are pleased to inform you that your manuscript 'Effectiveness of a multifaceted prevention programme for melioidosis in diabetics (PREMEL): a stepped-wedge cluster-randomised controlled trial' has been provisionally accepted for publication in PLOS Neglected Tropical Diseases.

Best regards,

Yunn-Hwen Gan

Associate Editor

Anna Ralph

Deputy Editor

---

## [Editor Report · Acceptance letter]

18 Jun 2021

Dear Dr. Limmathurotsakul,

We are delighted to inform you that your manuscript, "Effectiveness of a multifaceted prevention programme for melioidosis in diabetics (PREMEL): a stepped-wedge cluster-randomised controlled trial," has been formally accepted for publication in PLOS Neglected Tropical Diseases.

Best regards,

Shaden Kamhawi

co-Editor-in-Chief

Paul Brindley

co-Editor-in-Chief
